# The Remarkable and Selective In Vitro Cytotoxicity of Synthesized Bola-Amphiphilic Nanovesicles on Etoposide-Sensitive and -Resistant Neuroblastoma Cells

**DOI:** 10.3390/nano14181505

**Published:** 2024-09-16

**Authors:** Silvana Alfei, Paolo Giannoni, Maria Grazia Signorello, Carola Torazza, Guendalina Zuccari, Constantinos M. Athanassopoulos, Cinzia Domenicotti, Barbara Marengo

**Affiliations:** 1Department of Pharmacy, University of Genoa, Viale Cembrano, 16148 Genoa, Italy; carola.torazza@unige.it (C.T.); guendalina.zuccari@unige.it (G.Z.); 2Department of Experimental Medicine (DIMES), University of Genova, Via Alberti L.B., 16132 Genoa, Italy; paolo.giannoni@unige.it (P.G.); cinzia.domenicotti@unige.it (C.D.); 3Biochemistry Laboratory, Department of Pharmacy, University of Genoa, Viale Benedetto XV 3, 16132 Genova, Italy; mariagrazia.signorello@unige.it; 4Laboratory of Experimental Therapies in Oncology, IRCCS Istituto Giannina Gaslini, Via G. Gaslini 5, 16147 Genoa, Italy; 5Department of Chemistry, Campus Rio Achaias, University of Patras, 26504 Rio, Greece; kath@chemistry.upatras.gr; 6IRCCS Ospedale Policlinico San Martino, 16132 Genova, Italy

**Keywords:** high-risk neuroblastoma (HR-NB), HTLA-230 NB and HTLA-ER cells, triphenyl phosphonium groups, mitochondrial targets, bola-amphiphiles, nanosized vesicles

## Abstract

Neuroblastoma (NB) is a solid tumor occurring in infancy and childhood. Its high-risk form has currently a survival rate <50%, despite aggressive treatments. This worrying scenario is worsened by drug-induced secondary tumorigenesis and the emergency of drug resistance, calling for the urgent development of new extra-genomic treatments. Triphenyl phosphonium salts (TPPs) are mitochondria-targeting compounds that exert anticancer effects, impair mitochondria functions, and damage DNA at the same time. Despite several biochemical applications, TPP-based bola-amphiphiles self-assembling nanoparticles (NPs) in water have never been tested as antitumor agents. Here, with the aim of developing new antitumor devices to also counteract resistant forms of HR-NB, the anticancer effects of a TPP-based bola-amphiphile molecule have been investigated in vitro for the first time. To this end, we considered the previously synthesized and characterized sterically hindered quaternary phosphonium salt (BPPB). It embodies both the characteristics of mitochondria-targeting compounds and those of bola-amphiphiles. The anticancer effects of BPPB were assessed against HTLA-230 human stage-IV NB cells and their counterpart, which is resistant to etoposide (ETO), doxorubicin (DOX), and many other therapeutics (HTLA-ER). Very low IC_50_ values of 0.2 µM on HTLA-230 and 1.1 µM on HTLA-ER (538-fold lower than that of ETO) were already determined after 24 h of treatment. The very low cell viability observed after 24 h did not significantly differ from that observed for the longest exposure timing. The putative future inclusion of BPPB in a chemotherapeutic cocktail for HR-NB was assessed by investigating in vitro its cytotoxic effects against mammalian cell lines. These included monkey kidney cells (Cos-7, IC_50_ = 4.9 µM), human hepatic cells (HepG2, IC_50_ = 9.6 µM), a lung-derived fibroblast cell line (MRC-5, IC_50_ = 2.8 µM), and red blood cells (RBCs, IC_50_ = 14.9 µM). Appreciable to very high selectivity indexes (SIs) have been determined after 24 h treatments (SIs = 2.5–74.6), which provided evidence that both NB cell populations were already fully exterminated. These in vitro results pave the way for future investigations of BPPB on animal models and upon confirmation for the possible development of BPPB as a novel therapeutic to treat MDR HR-NB cells.

## 1. Introduction

### 1.1. Neuroblastoma (NB)

Neuroblastoma (NB) is one of the most common solid tumors occurring in infancy and childhood, but it is extremely rare in adults [1]. Precisely 90% of NB occurs in children under the age of 10, with most cases being diagnosed before five years of age [2]. NB causes 15% of all pediatric cancer deaths worldwide [3]. Usually, survival rates at younger ages are better than those at older ones [4]. NB is described as a heterogeneous developmental tumor, which can be caused by varying and different genes or alleles [5]. NB arises from the embryonic sympathoadrenal cells of the neural crest [5]. NB is one of the so-called enigmatic tumors [6] due to its unique and unpredictable clinical outcomes [4]. NB can evolve in spontaneous regression or with tumor maturation to benign ganglioneuroma, but it can also rapidly progress to life-threatening diseases and aggressive forms of cancer that are refractory to therapy [6,7]. In infants under 18 months of age, the disease often regresses spontaneously without therapy, whereas children over 5 years old with unfavorable histology tend to have poor outcomes [8]. Other cancers, such as lung tumors, are associated with environmental factors (asbestos and malignant mesothelioma), lifestyle behaviors, and smoking [9]. On the contrary, although certain exposures have been related to its onset in children [7], NB seems more closely associated with genetic and/or molecular characteristics [8]. NB can be classified as a low- and intermediate-risk disease, for which survival rates can range from 85 to 90% [10], as well as a high-risk disease (HR-NB) [4]. Unfortunately, about half of NB patients are diagnosed with HR-NB, which has a survival rate of 40–50% despite aggressive treatments [3]. In the 1980s, it was established that the onset of HR-NB with consequent low patient survival is strictly associated with MYCN gene amplification. From then on, several other genetic abnormalities have been identified in HR-NB, including gains of whole chromosomes and large-scale chromosomal imbalances [11]. Four stages (S1–S4), plus an additional special fourth stage (S4s), are recognized in the development of NB, where S4 and S4s the metastatic ones. However, stage S4s encompasses tumors with metastases confined to the skin, liver, and bone marrow, arising in children younger than 18 months of age [8]. While stages S1, S2, S3, and S4s of NB are characterized by a favorable histology that allows a survival rate in the range 85–90% [11], the metastatic stage S4 (NBS4) is often fatal despite aggressive therapy. Collectively, a survival rate in the range of 40–50% is accounted for by patients with NBS4 [12] due to the unfavorable histology of NBS4 [1], as previously reported for HR-NB. Treatments for NB generally consist of induction, consolidation, and maintenance therapy. In particular, HR-NB is initially treated with multi-agent chemotherapy, surgery, and radiotherapy [4]. Consolidation therapy consists of high-dose chemotherapy and autologous stem cell transplant (ASCT), often followed by further radiotherapy. Isotretinoin is instead commonly adopted as a maintenance therapy [13]. Recently, the treatment of NB has included anti-G2-immunotherapy by using monoclonal antibodies, such as dinutuximab or naxitamab, in combination with cytokines (GM-CSF and IL-2) and therapeutic agents [13]. Unfortunately, a retrospective study on this therapy has evidenced that ASCT did not provide survival benefit when anti-GD2 immunotherapy was used after induction chemotherapy [13]. Despite the addition of monoclonal antibodies, the survival rate for NBS4 remained at 40–50% [14], thus indicating that other molecular targets are urgently needed to efficiently treat NBS4. In fact, even though outcomes for patients with HR-NB have markedly improved over the past 2 decades, the available current therapies for HR-NB patients remain suboptimal [6]. In addition, to find other possible target therapies for the treatment of NBS4 patients, future medicinal chemistry research on NB should consider and adopt a multi-target drug approach, rather than a multi-drug–one-target one, to gain improved efficacy and fewer drug–drug interactions [4].

### 1.2. Possible Extra-Genomic Therapeutic Approaches

The chemotherapy of malignant neoplasms, including high-risk neuroblastoma (HR-NB), often leads to drug-induced secondary tumorigenesis due to nuclear DNA damage caused by traditional genotoxic chemotherapeutic agents. Furthermore, such diseases are commonly worsened by the emergency of drug resistance due to adaptive genetic mechanisms of cancer cells, including drug inactivation, altered drug targets, adaptive responses, and dysfunctional apoptosis [15].

In this regard, the use of a combinatorial treatment approach could delay the onset of resistance and improve patient outcomes. In any case, undesired drug–drug interactions could negatively affect the therapeutic effects [4]. Therefore, the search for compounds with an extra-genomic mechanism of antitumor action is of paramount importance in order to avoid the possible recurrence of cancer.

Recently, oxidative therapies (OTs), working via reactive oxygen species (ROS) induction by different methods [16,17], have been reviewed and reported as effective and alternative extra-genomic treatments for curing severe skin infections sustained by pathogens, including biofilm-producing microorganisms [18], without a tendency to develop resistance [17]. Similarly, high levels of oxidative stress (OS) caused by ROS induction have been reported as a novel type of anticancer therapy also capable of modulating the antitumor immune response without prompting resistance [19]. Specifically, photodynamic therapy (PDT) is a promising method of tumor ablation and function-preserving oncological intervention based on ROS induction by proper light-activated photosensitizers in the presence of oxygen [17]. It is described as minimally invasive and repeatable, with low side effects and no cumulative toxicity [20]. The combination of PDT with immune stimulation has been reported as a possible key to overcoming the melanoma resistance and to obtain better, sustainable clinical results [20]. Additionally, Watanabe et al. have demonstrated that PDT, based on 5-aminolevulinic acid as a photosensitizer agent, could be a new diagnostic, therapeutic, and surgical aid for neuroblastoma, triggering persistent apoptotic cell death [21].

Treatments involving ROS induced by PDT as antitumor agents are already clinically approved and have been widely employed against various tumors onto which irradiation can be applied directly, such as lung, esophageal, gastric, breast, head and neck, bladder, and prostate carcinomas [22]. In any case, some unavoidable disadvantages, including limited light penetration depth, poor tumor selectivity, and oxygen dependence, largely limit PDT therapeutic efficiency for the treatment of solid tumors, such as HR-NB, the object of the present study [22]. Moreover, further in-depth clinical investigations are mandatory to assess the possible toxic effects of extensive ROS administration to mammalian cells, biosafety, and target specificity [23,24].

So, the discovery of new anticancer molecules, acting via extra-genomic mechanisms, remains urgent. Within this context, mitochondria are the most promising target for an effective anticancer therapy due to their key role in energy production, apoptosis induction, and ROS generation [1]. Recent studies have shown that several cancer cells, including metastatic, therapy-resistant, and cancer stem cells, are reliant on mitochondrial respiration and upregulate oxidative phosphorylation (OXPHOS) activity to maintain tumorigenesis [25]. Mitochondria are crucial for tumor proliferation, survival, metastasis, and the development of resistance to chemotherapy and radiotherapy. Recent studies report that elevated heme synthesis and uptake lead to an intensification of mitochondrial respiration and ATP generation, thus promoting tumorigenic functions in non-small-cell lung cancer (NSCLC) cells and rendering mitochondria a vulnerable target for cancer therapy [25]. Very interestingly, these intracellular organelles possess their own DNA, independently and not associated with the mutable genetic mechanisms of cells [26]. Therefore, mitochondria-targeting compounds that are able to accumulate inside this organelle and to impair its functions would succeed in bypassing the genetic mechanisms underlying tumor recurrence and resistance, thus also being active in resistive cells. It has been reported that the triphenyl phosphonium (TPP) moiety is the most widely used mitochondria-targeting carrier [27]. Mono and *bis* TPP salts, due to the “soft” cationic nature of their headgroup(s), characterized by an extended delocalization of the positive charge on the phenyl rings, have a strong ability to interact and cross cell membranes, especially negatively hyper-polarized ones, compared to those of tumor cells [28]. Once they target the cancer cytoplasmic membrane, TTP salts can damage the phospholipid bilayer, causing depolarization and permeabilization and thus facilitating their entrance into the cell [29]. When inside the cancer cell, TPP-based compounds target mitochondria and interact and cross their double (outer and inner) membranes, which are even more negatively charged than the plasma membrane, thus causing their depolarization and/or destruction as well [30]. Mitochondria are the only intercellular organelles with a negative charge inside them. The ΔΨm between the matrix and the intermembrane space is around 180 mV [31]. Mitochondria use this potential as a proton-motive force to drive ATP synthesis, and ΔΨm is a key parameter that indicates the bioenergetic competence of mitochondria [32]. On the other hand, this ΔΨm is exploited for mitochondrial targeting by TPP salts and other cationic molecules that are primarily attracted to the mitochondria via an electrostatic force and then can amass inside them with 100- to 1000-fold accumulation [31].

The consequent inhibitory activity on the mitochondria functions determines mitochondrial toxicity and permanent damage to the organelle, thus causing the programmed death of cancer cells by the signal induction of cell death processes (apoptosis, necroptosis, or autophagy) [15,29,30]. Due to the strong difference in the negative electric membrane potentials of both cytoplasmic and mitochondrial membranes between tumor and normal cells, these compounds can selectively accumulate in the mitochondria of cancer cells, rather than in those of normal cells [33], thus resulting in low levels of cytotoxicity [28].

Bola-amphiphiles (BAs) are a particular class of cationic surfactants featuring one or more hydrophobic chains connecting two identical or different hydrophilic headgroups endowed with nonpareil colloidal properties [29]. They exhibit unique, hierarchically self-assembled structures, characterized by high polymorphism, and thus, they are attractive for applications in various fields, including drug delivery, gene delivery, electronics, and medical imaging [34]. *Bis*-triphenyl phosphonium (BTPP)-based bola-amphiphiles (BTPP-BAs), due to the presence of TPP groups, have been reported to be mitochondria-targeting compounds, with a high ability to cross the mitochondrial membranes and accumulate inside them [29]. A series of BTPP-BAs, featuring chains of 12, 16, 20, and 30 methylene units, have been studied for their colloidal properties, evidencing their capability to form nanosized vesicles and aggregates in water solutions [29]. It has been reported that BTPP-BAs can exert potent antibacterial properties and low toxicity in eukaryotic cells by mechanisms like those described above for non-bola-amphiphile TPP-based salts on cancer cells [35]. In any case, in the case of BTPP-BAs, it seems that their colloidal properties may also play a central role in their mitochondrial toxicity [29], suggesting an enhancement of antitumor effects with respect to traditional TPP salts. In this regard, we have recently reported the potent broad-spectrum antibacterial activity of a BTPP-BA molecule (BPPB) [35], which was remarkably more active than a previously prepared mono triphenyl phosphonium salt [36]. Alkyl triphenyl phosphonium salts [36,37,38,39], pegylated *bis*-triphenyl phosphonium salts [40], and BTPP-BAs [35,41,42] have been successfully tested as antibacterial agents. On the contrary, while conventional alkyl TPP-based salts have also been extensively studied as anticancer molecules [43,44,45,46], to our knowledge, while several reports exist on the several other applications of BAs, such as efficacious drug delivery systems loaded with antitumor therapeutics, no article exists on their use as anticancer agents per se [34].

### 1.3. The Present Study

Here, in the search for alternative molecules to counteract resistant forms of NB, the anticancer effects of a BTPP-BA *bis*-cationic molecule (BPPB) have been investigated for the first time. In particular, knowing that membrane-active cationic molecules that are effective as antibacterial agents are also often effective as antitumor agents, due to the similar membrane-negative constituents and composition of the two types of cells [47], we selected for this study the water-soluble sterically hindered quaternary phosphonium bola-amphiphile (BPPB) shown in Figure 1, featuring two triphenyl phosphonium groups linked by a C12 alkyl chain.

In fact, its broad-spectrum antibacterial effects on several MDR bacterial species by a membrane-active, not-specific, and extra-genomic mechanism have been recently reported [35]. Additionally, BPPB has been demonstrated to be very potent and selective for bacteria, while respectful of eukaryotic cells, including human hepatic and monkey kidney cell lines, and thus is very promising to be clinically developed in the future. In particular, it displayed selectivity indices >10 for all the 22 Gram-positive isolates testedand for clinically relevant Gram-negative superbugs such as those of *E. coli* species [35]. Then, aiming at extra-genetic anticancer effects mainly based on mitochondrial impairments, the selected *bis*-triphenyl phosphonium molecule appeared to be the best candidate for our scope, bearing both the mitochondria-targeting TPP groups and the peculiar colloidal properties of bola-amphiphiles reported to play a central role in their mitochondrial toxicity. As our interest was in finding new molecules that are also effective in the case of multidrug resistance, and since it was the first time that compounds like BPPB were tested as anticancer devices, before animal models were used, BPPB was assayed in vitro. To this end, BPPB was administered to sensitive NB cells (HTLA-230) and HTLA-ER NB cells, whose resistance to etoposide (ETO), doxorubicin (DOX), carboplatin, cis-platin, vincristine, etc., has been reported [48]. The excellent results observed on both cell populations stimulated us to assess the effects of BPPB administration toward different non-NB mammalian cells, envisaging its development as a novel agent suitable for clinical application. To this end, experiments were performed in vitro using human hepatic, monkey kidney, stromal lung cells, and red blood cells to obtain a complete overview of the possible toxic effects that BPPB could have on the main organs responsible for the disposal and circulation of drugs (liver, kidney, and blood) and on the extracellular matrix in the lungs.

## 2. Materials and Methods

### 2.1. Chemicals and Instruments

1,1-(1,12-dodecanediyl)bis [1,1,1]-triphenylphosphonium di-bromide (BPPB) was synthetized and characterized as recently described [35]. All analyses performed to characterize BPPB were carried out on instrumentation and with procedures that were previously reported [35].

### 2.2. In Vitro BPPB Cytotoxicity Evaluation on NB Cells

#### 2.2.1. Cell Culture Conditions

HTLA-230 human stage-IV NB cells were kindly provided by Dr. L. Raffaghello (G. Gaslini Institute, Genoa, Italy). The HTLA-ER cells were selected by treating HTLA-230 parental cells for six months with increasing concentrations of etoposide, as previously reported [49]. Both cell populations were maintained in RPMI 1640 medium (Euroclone Spa, Pavia, Italy) supplemented with 10% fetal bovine serum (FBS, Euroclone Spa, Pavia, Italy), 1% L-glutamine (Euroclone Spa, Pavia, Italy), and 1% Penicillin/Streptomicin (Euroclone Spa, Pavia, Italy) and grown in standard conditions (37 °C humidified incubator with 5% CO_2_).

#### 2.2.2. Treatments

To determine the cytotoxic effects of BPPB, in vitro time- and dose-dependent experiments were carried out. Cells were first treated for 24, 48, and 72 h, with increasing concentrations (0.5–50 µM) of the compound. In a subsequent series of the experiments, cells were treated for 24, 48, and 72 h with increasing concentration of BPPB in the range of 0.1–2 µM. The stock solutions of these compounds were prepared in 40,000-fold diluted DMSO, and pilot experiments demonstrated that the final DMSO concentrations did not change any of the cell responses analyzed. Cell cultures were carefully monitored before and during the experiments to ensure optimal cell density. Notably, samples were discarded if cell confluence reached >90%.

#### 2.2.3. Cell Viability Assay

Cell viability was determined by using the CellTiter 96^®^ AQueous One Solution Cell Proliferation Assay (Promega, Madison, WI, USA), as previously described [50,51]. Briefly, cells (10,000 cells/well) were seeded into 96-well plates (Corning Incorporated, Corning, NY, USA) and then treated. Next, cells were incubated with 20 µL of CellTiter, and the absorbance at 490 nm was recorded using a microplate reader (EL-808, BIO-TEK Instruments Inc., Winooski, VT, USA). IC_50_ was evaluated by GraphPad Prism 5.4.2 Software (GraphPad Software, Boston, MA, USA).

#### 2.2.4. Statistical Analyses

Results are expressed as the means ± S.D. of at least four independent experiments in which six different wells were analyzed every time for each experimental condition. The statistical significance of differences was determined by two-way analysis of variances (ANOVA) followed by Tukey’s multicomparison test using GraphPad Prism 8.0 (GraphPad Software v8.0, San Diego, CA, USA); *p* < 0.05 was considered statistically significant. Asterisks or other indicators (see Figures captions) indicate the following *p*-value ranges: *p* > 0.05 no symbols, * = *p* < 0.05, ** = *p* < 0.01, *** = *p* < 0.001, **** = *p* < 0.0001.

### 2.3. In Vitro BPPB Cytotoxicity Evaluation on Human Lung Fibroblasts MRC-5

#### 2.3.1. Cell Seeding Procedures and Treatments

Human MRC-5 lung fibroblast cells were used (ICLC-IRCCS San Martino, Genoa, Italy); cells were expanded in DMEM/F12 50%/50% (Dulbecco’s Modified Eagle Medium/Ham’s F12 50/50 Mix; Corning, Pisa, Italy; cat. n. 10-090-CV; ThermoFisher Scientific, Segrate (MI), Italy, supplemented with 10% fetal bovine serum (GIBCO-Invitrogen, GIBCO-ThermoFisher Life Science, Segrate, Italy, catalog number (cat. n.) 10270-098) and 2 mM glutamine (Euro-Clone; Pero, MI, Italy; cat. n. EC B3000D) at 37 °C under 5% CO_2_ aeration. After expansion, cells were washed three times with sterile phosphate buffered solution (PBS; VWR, Milano, Italy, cat. n. 392-0442), detached with 0.25% Trypsin solution (Corning; cat-25-050-CI), collected in complete medium, and centrifuged at 400× *g* for 12 min. The supernatant was discarded, and cells were suspended in PBS, counted, and centrifuged again as mentioned above. Cells were then resuspended in complete medium, seeded into 24-well plates (20,000 cells/well) and allowed to adhere to the plate for the next 24 h. Subsequently, non-adherent cells and the plating medium were discarded, and fresh standard medium or treatment medium were added. Treatment media contained BPPB at different concentrations (0.5, 1.0, 5.0, 25.0, 50.0, 75.0 and 100.0 μM); triplicate wells for each concentration were tested simultaneously, as well as controls. The medium was changed every day. Cells were photographed and assayed for viability prior to treatment (time 0; T_0_) and after 24, 48, and 72 h subsequently (T_24_, T_48_ and T_72_). Bright field images of each well/treatment were acquired using a Nikon Digital Sigh DS-5Mc camera (Nikon Europe B.V., Amstelveen, The Netherlands) mounted on an inverted Olympus CKX-41 microscope (Evident Europe GmbH, Segrate (Milan), Italy) (4×, 10× and 20× enlargements).

#### 2.3.2. Viability Test

The effects of BPPB on cell viability were assessed by the Presto Blue^TM^ assay (Invitrogen Presto Blue ^TM^ Cell Viability Reagent, cat. n. A13262, ThermoFisher, Monza, Italy), as suggested by the manufacturer. Briefly, at the indicated time points, cells were washed twice with PBS and incubated in the dark a 37 °C for 4 h with fresh complete medium supplemented with 10% Presto Blue ^TM^. Aliquots of the Presto Blue ^TM^-containing medium were incubated in the same conditions, to be used as reaction blanks. Supernatants and blanks were then collected and analyzed using a Spectra MR multi-plate reader spectrophotometer (Dynex Technologies; Chantilly, VA, USA) at 570 and 600 nm. The results were calculated as the average of treatment replicas (triplicates) ± S.D., normalizing each 570 nm reading to its corresponding value at 600 nm, subtracting the average blank value, and normalizing the results at each time point to the initial reading at T_0_.

### 2.4. In Vitro Concentration-Dependent Cytotoxicity Experiments on Other Mammalian Cells

#### 2.4.1. MTT Cytotoxicity Assay on Cos-7 and HepG2 Cells

To assess the cytotoxicity properties of BPPB, the MTT (3-(4,5-dimethylthiazol-2-yl)-2,5-diphenyltetrazolium bromide) cell proliferation assay [52] (Abcam, Prodotti Gianni s.r.l., Milano, Italy, Cat#ab2011091) was performed, following the manufacturer’s protocol, as previously reported [35,36]. This time, the increasing concentrations of BPPB were in the range of 0.4–85.3 µM. The cell survival rate, expressed as cell viability percentage (%), was evaluated based on the experimental outputs of treated groups vs. the untreated groups (CTR) and was calculated as follows: cell viability (%) = (OD treated cells − OD blank)/(OD untreated cells − OD blank) × 100%.

#### 2.4.2. LDH Cytotoxicity Assay on Cos-7 and HepG2 Cells

To assess the cytotoxicity properties of BPPB, the lactate dehydrogenase (LDH) cytotoxicity assay [53] (Abcam, Cat#ab102526) was performed, following manufacturer’s protocol, as previously described [35]. This time, the increasing concentrations of BPPB were in the range 0.4–85.3 µM.

#### 2.4.3. Statistics

Graphs and statistics were generated by GraphPad Prism (Version 9, license code GP9-2314983-RATL-05225; 225 Franklin Street. Fl. 26, Boston, MA, USA 02110; RRID:SCR_002798).

### 2.5. In Vitro Hemolytic Toxicity of BPPB Using Red Blood Cells (RBCs)

#### Hemolysis

The hemolytic ratio was evaluated, as recently reported with slight changes [54], in EDTA-blood samples from four healthy donors from the San Martino Hospital Transfusion Centre. Specifically, red blood cells (RBCs) were isolated by diluting 0.2 mL of blood with 0.4 mL of phosphate-buffered saline (PBS) and centrifuging it for 5 min at 10,000× *g*. The pellet, consisting of RBCs, was washed 5 times with 1.0 mL PBS and finally resuspended with 2.0 mL PBS. The assay was carried out on 0.1 mL of resuspended RBCs added to 0.1 mL of H_2_O (positive control), 0.1 mL of PBS (negative control), or 0.1 mL of the different concentrations (0.1–50 µM) of the compound to be tested (BPPB). The samples were incubated for 60 min at 37 °C, and at the end, they were centrifuged for 5 min at 10,000× *g*. Finally, 0.1 mL of supernatant was transferred to a transparent 96-well plate and measured by a spectrophotometer (Dynex Technologies; Chantilly, VA, U.S.A.) at 595 nm. The hemolytic ratio was calculated using the following formula:Hemolytic ratio (%) = (OD _TEST_ − OD _NEGATIVE CONTROL_)/(OD _POSITIVE CONTROL_ − OD _NEGATIVE CONTROL_) × 100

## 3. Results and Discussion

### 3.1. 1,1-(1,12-dodecanediyl)bis [1,1,1]-triphenylphosphonium di-Bromide (BPPB)

With the aim of developing new antitumor agents effective against high-risk nuroblastoma (HR-NB), the alkyl triphenilphosphonium derivative displaying two triphenyl cationic moieties linked to each other by a C12 alkyl chain (BPPB, Figure 1) was synthetized according to Figure 1, according to the recently described procedure [35].

As reported, molecules possessing the TPP group can exert antibacterial and antitumor effects due to their easy crossing and damaging of malignant bacterial and eukariotic cell membranes. In the case of tumour cells, upon the selective electrostatic interaction with the negatively charged constituents of their cytoplasmic membrane, such compounds enter the cancer cell and target mitochondria, into which they accumulate, impairing the essential functions for cell survival and causing cell death [29,55,56]. In this regard, although bola-amphiphilic structures have never been tested as anticancer agents, we were confident that the two TPP groups could drive the accumulation of BPPB into cancer cell mitochondria. Once inside these organelles, BPPB could disrupt the mitochondrial membrane potential, which is essential for the production of ATP exerting mitochondrial toxicity, and cause the programmed death of recipient cancer cells by the induction of specific processes (apoptosis, necroptosis or autophagy) [15,29,30]. In addition, the peculiar bola-amphiphilic structure, conferring BPPB nonpareil colloidal properties and the capacity to self-assemble in a water solution in nanosized spherical vesicles and to generate nanoaggregates, could further increase its mitochondrial toxicity, as reported by Ceccacci et al. [29].

#### BPPB Characterization

Associated with a numbered structure of BPPB (Appendix A in Appendix A), the list of peaks from ATR-FTIR, ^1^H, ^13^C and ^31^P NMR spectra, as well as results from FIA-MS-(ESI) and elemental analyses, are included in Appendix A. Other useful information on previously assessed BPPB characteristics, significant images, and biological effects are included in the Appendix A [35].

### 3.2. In Vitro Concentration- and Time-Dependent Cytotoxic Effects of BPPB on Neuroblastoma (NB) Cells

We evaluated the cytotoxicity of BPPB on HTLA-230, an ETO-sensitive NB cell line, and HTLA-ER, the counterpart resistant to several chemotherapeutics that are currently clinically used, including etoposide (ETO), doxorubicin (DOX), carboplatin, cis-platin, and vincristine [48]. The two cell populations were exposed to BPPB in an explorative concentration range of 0.5–50 µM for 24, 48, and 72 h to detect the lower range of cytotoxic concentrations. The concentration = 0 µM, i.e., cells not treated, was assumed as the control (Ctr).

#### In Vitro Concentration- and Time-Dependent Cytotoxic Effects of BPPB on NB Cells

As reported in Figure 2 (HTLA-230 cells) and Figure 3 (HTLA-ER cells), BPPB exerted mainly dose-dependent cytotoxic effects, and its toxicity was less dependent on the time of exposure. In any case, for concentrations ≥ 5 µM (HTLA-230) and ≥10–15 µM (HTLA-ER), cell viability was very low in both populations and practically constant for all other concentrations of BPPB tested. In particular, while at concentrations = 0.5 µM, the percentage of viable HTLA-230 cells was already under 50%, and an effect of exposure time was observed (viable cells were 46.0% after 24 h, 41.6% after 48 h, and 21.7% after 72 h of exposure); at concentrations ≥ 5 µM, viable cells were constantly in the range of 16–18%.

In the case of HTLA-ER NB cells, cell viability at BPPB concentrations = 0.5 µM was strongly affected by the time of exposure. In particular, while, after 24 h of treatment, cell viability was not significantly different from that of controls, it was shown to be significantly reduced both after 48 h (54.1%) and 72 h of exposure (62.0%), evidencing a higher tolerance of these cells with respect to HTLA-230. For concentrations ≥ 5 µM, the effect of exposure time was minimal, and although at BPPB concentration = 5–10 µM, a slight dose-dependent effect was observed at all exposure times, a plateau in the viability of cells was detected for all other concentrations tested with very low percentages of viable cells (7–8%), even lower than those observed on sensitive HTLA-230. Collectively, these early experiments evidenced that for concentrations ≥ 5 µM, NB cells were completely eradicated. With these results, we calculated the IC_50_ of BPPB on both cell lines, using GraphPad Prism software (version 8.0.1) and fitting data with non-linear regression models. In particular, we first converted the bar graphs in Figure 2 and Figure 3 in dispersion graphs (Appendix A); then, upon conversion of µM concentrations (x) in Log_10_ (x) and using a non-linear model that considered the Log_10_ (BPPB concentrations) vs. the normalized response, we derived the IC_50_ values of BPPB for both cell populations at 24, 48, and 72 h of treatment reported in Table 1.

To refine the data reported in Table 1, we restricted the range of concentrations to be investigated, and those over 2 µM were not further considered. In this new series of experiments, both NB cell populations were exposed to increasing concentrations (0.1–2 µM) of BPPB for 24, 48, and 72 h, and the effects of these treatments on cell viability were evaluated and are reported in Figure 4 and Figure 5.

As depicted in Figure 4, in the case of HTLA-230 NB cells, a significant reduction in cell viability was observed for concentrations ≥ 0.1 µM at all time points. For concentrations = 0.25 µM, an effect of the duration of exposure was observable between 24 and 48 h, with cell viability of 53.1 and 41.6%, respectively. On the contrary, for concentrations ≥ 0.5 µM, cell viability was lower after 24 h of treatment with respect to treatments for 48 and 72 h (22–15% vs. 23–17% vs. 34–18%). HTLA-ER NB cells demonstrated more tolerance to BPPB than HTLA-230 ones, and a significant reduction in viability (*p* < 0.0001) was observed for concentrations ≥ 0.5 µM. In any case, while a decrease in cell survival, dependent on both concentrations and times of treatment, was detected in the range of 0.5–1.25 µM, a scenario like that described above for HTLA-230 NB cells was observed for higher concentrations, evidencing a diminished cell viability after 24 h of treatment with respect to treatments for 48 and 72 h (33–14% vs. 39–31% vs. 36–28%). Collectively, these new observations suggest a substantial reduction in the previously reported values of IC_50_. Upon conversion of the bar graphs in Figure 4 and Figure 5 into dispersion graphs (Appendix A), and by applying the same method described above to generate the plots shown in Appendix A, we calculated and report in Table 2 the new IC_50_ values.

The data reported in Table 2 evidence that, as expected, refined IC_50_ values were significantly lower than those calculated from data from earlier experiments. Additionally, although very low IC_50_ values were also achieved on HTLA-ER NB cells (0.8–1.1 µM), these cells were more tolerant to BPPB than their HTLA-230 counterparts, with a very small dependence on the time of exposure (1.1 vs. 0.9 vs. 0.8 after 24, 48, 72 h treatment). On the contrary, HTLA-230 NB cells demonstrated very low (0.2 µM) IC_50_ values, fully independently of the time of treatment. Collectively, very low IC_50_ concentrations (0.2–1.1 µM) were sufficient to fully exterminate both cell populations after only a 24 h treatment. Despite the design, synthesis, and formulation of novel compounds, polymers, and dendrimers, in most cases nano-dimensioned, finalized to treat resistant forms of cancer, including HR-NB, have represented one of our most important challenges for years, this is the first time that we observe values of IC_50_ so low on NB cells, and with no need to use particular and/or additional formulation techniques. Against NB and using HTLA-230 and HTLA-ER NB cells as in this study, we prepared and assayed nanosized copolymers (namely P5 and P7) in 2021, observing IC_50_ = 4.3 µM (P5) and 5.1 µM (P7) against HTLA-230 and 2.2 µM (P5) 4.1 µM (P7) against HTLA-ER [57]. More recently (2023), we assayed an imidazo-pyrazole derivative against both cell populations, finding modest antiproliferative effects [58]. However, being already more active than etoposide against HTLA-ER after 24 h of exposure, we have formulated it as nanoparticles (NPs), using palmitic acid and P5 of the previous work [57], thus increasing further its anticancer effects especially after 72 h of exposure (8.62 µM on HTLA-230 and 13.55 µM on HTLA-ER) [58]. In 2020, we also prepared a fifth-generation polyester-based dendrimer with a propanediol *core* and bearing 64 peripheral hydroxyls (namely 4) and used it to encapsulate etoposide (ETO), with the scope of reducing its degradability and/or improving its solubility and antiproliferative effects on HTLA-230 NB cells [50]. Unexpectedly, we found that 4 possessed an intrinsic, even if modest, antitumor effect when administered alone at the low concentration of 0.169 µM. In any case, it was capable of reducing HTLA-230 cells viability by only 40%. More interesting, it was revealed to be an excellent delivery system for ETO, and when HTLA-230 NB cells were treated with etoposide-enriched 4 for 72 h, their viability was reduced by 65% with respect to untreated cells and by 30% with respect to cells treated with the same amount of free ETO as that released by the dendrimer [50]. Other attempts to further improve the anticancer effects of 4 via its covalent bond with gallic acid (GA), or by entrapping GA in its cavities, only enhanced its cytoprotective effects, making it a promising adjuvant treatment in the early stage of NB or for its prevention [51]. Our best achievement was obtained in 2022, when copolymer P7 from our previous work [57] was used to formulate a pyrazole derivative that waws previously reported to have moderate antibacterial properties in nanoparticles (CB1H-P7 NPs), which had demonstrated significantly enhanced antibacterial effects [59]. In this case, the IMR-32 and SHSY 5Y cell lines were used as cell models of NB, while HaCaT human keratinocytes were selected to test CB1H-P7 cytotoxicity. Notably, the IC_50_ and SI values for CB1H-P7 NPs were 0.43–0.47 µM (dependent on the time of treatment) and 2.9–3.3 against IMR-32 cells, and 0.47–0.54 µM and 2.7–2.8 against SHSY 5Y cells. Comparing these results with those obtained using fenretinide (4-HPR), which has been reported as an antitumor compound under a phase III clinical trial and is also effective against tumors that are unresponsive to retinoic acid treatments, on the same cells, CB1H-P7 NPs was shown to be 1.6–14.5-fold more potent than 4-HPR, acting both through apoptosis, as 4-HPR does, and through necrosis [60]. In this context of already excellent results, on HTLA-230, with no pattern of resistance, like IMR-32 and SHSY 5Y NB cells, those obtained with BPPB were even more brilliant, showing BPPB 2–2.5-fold more potent than CB1H-P7 NPs. Even more importantly, the relevance of our results (at least in vitro) is derived from the very potent cytotoxicity of BPPB to NB cells if compared to that of etoposide (ETO), which is one of the most clinically used drug to treat HR-NB. As reported, the in vitro results demonstrated that, upon administration of 100 µM ETO for 24 h, i.e., a dose 91 and 500 times higher than that needed for BPPB to kill 50% of NB cells, the cell viability of HTLA-ER and HTLA-230 remained over 80% and close to 70%, respectively [49]. Also, in a more recent work, it was reported that the extrapolated IC_50_ of ETO on HTLA-ER was 592 µM, i.e., 538-fold higher than that of BPPB [58]. Since this is the first time that a TPP-based bola-amphiphile molecule is evaluated for its possible anticancer effects, the comparison of our results with others previously reported for molecules of the same class is currently impossible. In any case, based on the existing literature data, as reported in the Introduction—concerning the anticancer effects of TPP-based compounds acting as membrane disruptors and targeting mitochondria, as well as those concerning the capability of TPP-based bola-amphiphiles to target mitochondria as well, thus dramatically impairing their function—we can assume that BPPB could also act by such mechanisms. In particular, upon electrostatic interaction with the negatively charged cytoplasmic membrane of NB cells, BPPB would cause depolarization, thus priming cell death, creating pores and entering the cell. Once inside, it would interact with negative mitochondria membranes, cause depolarization and permeabilization, and enter mitochondria, impairing their functions, which are vital for the cells’ respiration and life, and inducing high ROS production and oxidative stress (OS), leading to DNA damage, lipid and protein peroxidation, and cell death. In this regard, although a study is underway where the location of BPPB once inside the cell, as well as its effects at molecular levels, will be assessed, the strong induction of ROS hyperproduction has been already confirmed.

### 3.3. Evaluation of the Effects of BPPB on Different Non-Tumoral Cells and Red Blood Cells

The assessment of cytotoxicity toward noon-tumoral mammalian cells is an essential step to predict if new compounds under investigation could be worthy of development as novel agents for clinical application or can serve as a template platform for the development of more active and less toxic derivatives by chemical transformations. As reported, the cytotoxicity of QPSs, including BPPB, could depend on their physicochemical characteristics, such as the length of the alkyl chains, the presence of aromatic ring(s) on which the P^+^ cation can delocalize [29,35], the type of counter anion, possible colloidal properties, and the capability or lack thereof to form nanoaggregates [61]. Additionally, QPS cytotoxicity could also depend on the specific cell type under investigation [61]. Concerning this latter aspect, cancer cells present differences from non-tumoral mammalian cells, including immortalized ones. The presence of lipids with a net anionic charge on the cancer cell’s surface, in place of zwitterionic lipids that are neutral at physiological pH, displayed on normal cells, translates into a significant disparity in the membrane’s electrostatic charges, resulting in the targeted tropism of cationic molecules toward the more negatively charged tumor cells [47]. Additionally, cholesterol, which is present to a larger extent in the membranes of normal cells than cancer ones, could provide selective protection for the first ones, thus reducing the possible cytotoxicity of a novel cationic compounds upon exposure [47]. In any case, mainly depending on the concentration, the time of exposure, and the specific structural properties of QPSs, cytotoxicity still also manifests in normal mammalian cells [61]. On these grounds, we investigated BPPB’s toxic effects on mammalian cell lines, including non-immortalized human MRC-5 lung fibroblasts, as well as immortalized human hepatic HepG2 cells and monkey kidney Cos-7 cells. Additionally, its hemolytic toxicity was also assessed on red blood cells (RBCs).

#### 3.3.1. Evaluation of the Concentration- and Time-Dependent Effects of BPPB on Human Lung Fibroblasts MRC-5

MRC-5 cells are a non-immortalized human lung fibroblast cell line derived from embryonic tissue and are commonly used in biomedical research [62]. Human stromal MRC-5 cells were seeded in complete medium into 24-well multi-well plates (20.000 cells/well) and allowed to adhere for 24 h. To evaluate the seeding uniformity, all wells were photographed before treatments (time zero, T_0_) at both low (4×) and medium (10×) magnification (Figure 6). Cells were then treated with different BPPB concentrations in the range of 0.5–100.0 μM. Three wells were assessed for each concentration tested, as well as for controls (CTRL; untreated cells).

At the indicated times of 24, 48, and 72 h, all wells (CTRL and BPPB-treated ones) and those treated with BPPB 0.5–100 µM were photographed again before the cell viability assessment (images available in Appendix A); cell viability was evaluated by the Presto Blue assay (a vital dye that provides color proportionality to the mitochondrial activity, i.e., to the number of metabolically active cells) and measuring the absorbance (Abs) at 570/600 nm. The measurements were performed in duplicate for each well and in triplicate for each concentration tested at all selected time points. The results are expressed as the mean of independent experiments ± S.D. They are reported in Appendix A, both as a dispersion graph of the measured absorbance (Abs) vs. times of treatment (A) and as bar graph of BPPB concentrations (0–100 µM) vs. cell viability (%) (B). Figure 7 shows the bar graph resulting from plotting the BPPB concentrations in the range of 0–5 µM vs. cell viability (%) measured after 24, 48, and 72 h. The appearance (Appendix A) and viability (Appendix A) of the control cells are in line with a normal trend, which shows a slight physiological decline of cell survival within the first 24 h after plating and a prompt recovery of cell growth afterward. The morphology of control cells (CTRLs) remained unchanged even when the cells were examined at higher magnifications (Appendix A). Increasing the concentrations of BPPB caused alterations in cell morphology, starting with rounding, the appearance of vacuoles and blebs, and the detachment from the plate. Any BPPB concentration equal to or higher than 25 μM was sufficient to cause the detachment and death of all plated cells within 24 h (Appendix A). Consequently, no images of the effects of 50, 75, and 100 μM BPPB were shown for 48 h of treatment (Appendix A), while in Appendix A (72 h of treatment), only the images of the effects of BPPB up to 5 μM were reported. In fact, for 48–72 h of exposure, the effects on morphology and viability observed at 24 h for BPPB concentration >25 µM were also evidenced at lower BPPB concentrations (1.0 μM–5.0 μM BPPB). In particular, at the time points of 48 and 72 h, the percentage of viable cells at 1.0 μM BPPB was significantly lowered to 26.81% and 30.63% respectively, with respect to the seeded cells at T_0_ (100.0%; Appendix A), whereas it dropped to even lower values of 2.78% and 3.04%, respectively, at 5.0 μM BPPB.

These data strongly suggest that BPPB exposure could trigger the apoptotic onset in treated cells, possibly impairing the mitochondrial functionality and/or efficiency, due to its reported tropism for mitochondria because of the presence of TPP group in its structure [29,35]. Interestingly, it must be noted that the removal of BPBB from the culture medium allowed the post-treatment recovery of the surviving cells and the reprise of normal growth.

Figure 7 reports the viability of CTRL cells putatively treated with BPPB but at time zero (T_0_) (100%); their viability after 24, 48, and 72 h without BPPB administration; and cell viability after 24, 48, and 72 h of treatment with increasing concentration of BPPB. Concentrations > 5 µM are not reported in Figure 7 because cell viability was insignificant and close to 0, as indicated above.

In Figure 7, the fluctuation in the viability of untreated controls is clearly visible. Viable cells decreased from 100% to 92% after 24 h and to 85% after 48 h and promptly recovered their growth up to 138% after 72 h. Rising concentrations of BPPB caused an increasing reduction in cell viability that was strongly dependent on the time of treatments. Since at the highest concentration shown in Figure 7 (5 µM), the cell viability was 38% after 24 h of exposure, and it was only of 3% and constant for higher times of treatment (48–72 h), prolonged treatments with BPPB at concentrations > 0.5 µM are inadvisable.

To have data that are directly comparable with those obtained on NB cells, upon conversion of the bar graphs in Appendix A and Figure 7, in dispersion graphs, and by applying the same procedure reported in the previous sections, the IC_50_ values of BPPB were calculated in the range of concentrations of 0–100 and 0–5 µM, respectively, without observing significant differences. Appendix A report the plots of BPPB concentrations (0–100 and 0–5 µM) vs. cell viability (%) after 24, 48, and 72 h treatment, while Appendix A shows the transformed plots and nonlinear models used to obtain the IC_50_ values of BPPB on MRC-5 cells. The IC_50_ values of BPPB on MRC-5 cells when administered at concentrations in the range of 0.5–5 µM are presented in Table 3.

Considering the results obtained at 24 h, given that NB cell viability was already very low and did not change significantly for longer times of treatment, the IC_50_ values were remarkably higher than those measured on both HTLA-230 (13.9 times) and HTLA-ER NB (2.5 times) cells.

#### 3.3.2. Assessment of the BPPB-Concentration-Dependent Effects on Cos-7 and HepG2 Cells by MTT and LDH Assays

In this study, the cytotoxicity profile of BPPB was assessed in both immortalized human liver cells (HepG2) cells [63,64,65,66], as in our previous works [35,36], and in transformed African green monkey kidney fibroblast cells (Cos-7), as reported in a recent paper [35]. Specifically, we evaluated the cytotoxicity of BPPB by carrying out both the MTT proliferation assay and the lactate dehydrogenase (LHD) cytotoxicity assay, which measures the LDH activity as an indicator of cell death. Both cell lines were exposed for 24 h to increasing concentrations of BPPB (0.4–85.3 µM). Growth inhibition/death were determined by the MTT test, while cell cytotoxicity was determined by LDH assay, with respect to untreated control cells. Results are reported as bar graphs in Figure 8 and as dispersion graphs in Appendix A.

In particular, in the case of the MTT test (Figure 8a,b), bar graphs show the cell viability (%) of untreated Cos-7 (a) and HepG2 (b) control cells (CTRLs) and after exposure to increasing concentrations of BPPB for 24 h, while in Figure 8c,d, we plot the quantification of cell damage (expressed as optical density (OD) units of absorbance)) of CTRL Cos-7 (c) and HepG2 (d) cells and after exposure to increasing concentrations of BPPB for 24 h.

According to the results shown in Figure 8a,b, in the MTT test, BPPB demonstrated concentration-dependent effects toward both cell lines. In any case, for concentrations = 8.5 µM, while on HepG2 cells, BPPB exerted effects like those observed at 4.3 µM and cell viability was higher than 50% and similar (64.5% vs. 62.9%) at both concentrations, the cell viability of Cos-7 decreased under 30% (27.45%) with respect to that observed at 4.3 µM (64.2%). Collectively, the MTT test evidenced that the cell viability was affected by BPPB exposure to a more major extent in Cos-7 cells than in HepG2 ones; a safety profile was detected at concentrations ≤ 4.3 µM toward Cos-7 and ≤8.5 µM toward HepG2 cells, which is, however, a scenario that is more satisfactory than that recently reported by Nunes et al., for the best-performing antibacterial compounds they synthesized [61]. From the LDH test results (Figure 8c,d) a slightly different scenario was evidenced, in which, while a dose-dependent effect was observed at concentrations ≥ 8.5 µM on Cos-7 cells and similar cell damage was observed for concentrations ≤ 4.3 µM, on HepG2 cells, a dose-dependent effect was observed at concentrations ≤ 8.5 µM, and similar cell damage was observed for concentrations ≥ 42.6 µM. Nonetheless, as observed in the MTT test, for concentrations = 8.5 µM, while on HepG2 cells, BPPB caused damage like that observed at 4.3 µM and the optical density due to LDH release was similar (OD = 0.71 vs. 0.79) at both concentrations, the damage measured on Cos-7 cells remarkably increased (OD = 0.80) with respect to that observed at 4.3 µM (OD = 0.30), thus evidencing remarkably higher cell damage. Collectively, the LDH test also evidenced that a safety profile exists at concentrations ≤ 4.3 µM toward Cos-7 and ≤8.5 µM toward HepG2 cells. For a more precise evaluation of the effects of BPPB on the Cos-7 and HepG2 cells considered here and to compare them with those found on NB cells (targets to be eliminated) and on previously considered stromal cells (by-stander cells of the microenvironment to leave unaffected), as models of the different cell types co-existing in a putative target tumoral tissue, we calculated the IC_50_ of BPPB on both cell lines. The transformed plot of Log_10_ BPPB concentrations vs. cell viability (%) and the nonlinear models are shown in Appendix A in the Appendix A. The calculated IC_50_ values were 4.91 ± 0.81 µM and 9.64 ± 1.31 µM on Cos-7 and HepG2 cells, respectively (Table 4), thus confirming the significantly dissimilar cytotoxicity of BPPB on the different cells, where the MRC-5 stromal cells were the least, the Cos-7 the intermediate, and the HepG2 the most tolerant ones. These findings confirm that the toxicity of a new compound strongly depends on the type of cell line used [61].

The lower toxicity of BPPB found toward human hepatic cells with respect to monkey kidney ones (and not the contrary) is a very promising result, which could pave the way for the future clinical development of BPPB and its possible derivatives. Additionally, since it is believed that the HepG2 cell line retains most of the metabolic functions of normal hepatocytes, and it is commonly used to study the toxic effects of heavy metals, nanoparticles, and drugs in vitro [67], for our scope, the cytotoxicity results obtained on human HepG2 are more reliable than those obtained on monkey Cos-7 cells.

#### 3.3.3. In Vitro Hemolytic Toxicity of BPPB on Red Blood Cells (RBCs)

The hemolytic ratio percentage (%) caused by BPPB was assessed as recently reported with slight changes [54]. In particular, EDTA-blood samples from four healthy donors were exposed to increasing concentrations of BPPB (0.1–50 µM). The results are expressed as the means of the four independent determinations ± S.D. and are shown in Figure 9.

As is observable, hemolysis was statistically significant with respect to control (CTRL) only for concentrations ≥ 10 µM but limited to 25.1% for BPPB = 10 µM. A concentration of BPPB = 20 µM was necessary to determine a hemolysis over 50% (50.8%). Figure 10 shows a comparison between cancer cell viability and the viability of RBCs when exposed to the same concentrations of BPPB. For both HTLA-230 and HTLA-ER NB cells, the minimal treatment duration of 24 h was considered.

A dispersion graph showing the comparison between the viability of the same cell populations when exposed to all concentrations tested for single experiments is included in the Appendix A. Confirming the evidence already demonstrated by Figure 10, both populations of NB cells were fully exterminated (viability = 18.2% for HTLA-230 and =14.3% for HTLA-ER) when hemolysis was inexistent, and RBC viability was still 98.4%. The dispersion graph of BPPB concentrations vs. RBC viability (green trace in Appendix A) was converted as previously described in the dispersion graph of Log_10_ BPPB concentrations vs. RBC viability (green trace with indicators and error bars in Appendix A). Then, the plot of the nonlinear fit of Log_10_ concentrations of BPPB vs. the normalized response (green trace without indicators in Appendix A) was obtained and used to calculate the IC_50_ (intended as the concentration of BPPB needed to cause 50% hemolysis), which was 14.92 ± 10.80 µM. Before using the IC_50_ values obtained on all cancer and normal cells considered here to assess the selectivity of BPPB for NB cells in relation to its toxicity toward normal cells, we have utilized these data to make a bar graph comparing them, which is available in Appendix A.

### 3.4. Selectivity Indexes

The selectivity index (SI) was calculated to evaluate the toxicity of BPPB against non-tumoral cells and to predict its therapeutic potential. High SI values result from large differences between the cytotoxicity against cancer (NB, in our case) and non-tumoral cells (MRC-5, Cos-7, HepG2, and RBCs, in our case) and indicate that cancer cells will be killed at a higher rate than normal ones [68]. In particular, the SI values of BPPB were calculated in relation to all non-tumoral cell (NTC) populations considered and all times of treatments available using Formula (1):SI = IC_50_ for NTC/IC_50_ for NB(1)

According to literature and a recent article by Krzywik et al., an SI > 1.0 can be considered favorable for the development of a new molecule as chemotherapeutic [68].

#### 3.4.1. Selectivity Indices toward MRC-5, Cos-7, and HepG2 Cells

The SIs of BPPB toward NB cells in relation to its toxicity on MRC-5, Cos-7, and HepG2 cells are reported in Table 5 and in Figure 11.

As evidenced in Table 5 and Figure 11, after 24 h of treatment, the time at which BPPB had already exerted its maximum of toxicity and cell viability did not change further for longer times of exposure, and SIs >> 2 were found in relation to both HTLA-230 and resistant HTLA-ER NB cells. In particular, the lowest SI = 2.5 was determined on MRC-5 cells, followed by Cos-7 (SI = 4.42) and HepG2 cells (SI = 8.68), if multidrug-resistant HTLA-ER NB cells are considered. Very much higher SI values all >10 and up to 40.7 were instead found when HTLA-230 cells were considered. Exclusively for MRC-5, the cells which resulted less tolerant to BPPB, it was possible to calculate the SI values for longer periods of exposure. Nevertheless, these additional data are of limited importance since the viability of NB cells of both populations tested remained unchanged after 24 h of treatment. SI values ≈ 1, reported as a promising value by Krzywik et al. [68], were also found after 48 and 72 h of exposition for MDR HTLA-ER NB cells, while higher values of 3.77 and 4.01 were found at 48 and 72 h of exposition when sensitive HTLA-230 NB cells were considered. However, the apparently not exceptional SI of BPPB for MDR HTLA-ER NB cells, calculated considering its cytotoxicity vs. MRC-5 cells after 24 h treatment, was remarkably higher than that of ETO, a clinically approved drug to treat HR-NB, as calculated using the reported average IC_50_ value obtained by Iijima et al., from evaluations on three human normal oral cell lines [69] and the ETO IC_50_ value determined by us on the same NB cells, after 24 h of exposure [58]. Specifically, the SI of BPPB for MDR HTLA-ER NB cells (2.49) was 7.8-fold higher than that of ETO (0.32). Collectively, at least in vitro, BPPB was safe for all cells tested, and, in a future clinical administration, it may possibly be safe for organs such as kidney and liver for treatments lasting 24 h. The determined SI values, which also provide indications concerning the susceptibility/tolerance of non-tumoral cells toward a novel compound, were strongly dependent on the type of cells used. The high susceptibility of MRC-5 human lung fibroblasts with respect to HepG2 ones may be due to a smaller arsenal of detoxifying enzymes possessed by these cells with respect to hepatocytes. In fact, while the liver and hepatocytes are a highly specialized organ and cells, respectively, that play a pivotal role in the detoxification of numerous substances, the detoxifying capacity of fibroblasts (as MRC-5 cells) is indeed lower with respect to hepatocytes. While hepatocytes are responsible for many functions, including detoxification, with remarkable proliferation capabilities and playing a crucial role in liver homeostasis, repair, and regeneration [70], fibroblasts are more involved in the wound-healing process and the deposition of extracellular matrix during liver fibrosis [71]. All these reasons explain the greater tolerance of HepG2 cells compared to both human fibroblast MRC-5 and monkey kidney Cos-7 cells toward exogenous compounds such as our BPPB. Furthermore, the fact that, unlike Cos-7 and HepG2, MRC-5 cells are not immortalized makes them even more susceptible to the toxic action of BPPB. Krzywik et al. reported the SIs of several synthesized colchicine derivatives toward several cancer cell lines with respect to their cytotoxicity on murine embryonic fibroblast cells [68]. In particular, they reported SI values in the ranges 1.5–4.0, 3.0–4.8, 5.6–7.8, 6.5–9.1, 6.6, and 9.8 depending on both the structural characteristics of the tested compounds and on the type of cancer cells [68]. Although some of the reported SIs support the future development of their compounds for clinical applications, their safest compound (SI = 9.8) displayed a SI value that was 4.1-, 2.1-, and 1.2-fold lower than those of BPPB on HepG2, Cos-7, and MRC-5 cells, when its anticancer effects on HTLA-230 at 24 h of treatment were considered.

#### 3.4.2. Selectivity Indexes toward RBCs

The SIs of BPPB toward NB cells in relation to its hemolytic toxicity on RBCs are reported in Table 6 and in Figure 12.

Very high SIs (>74) were observed for HTLA-230 NB cells. In any case, for MDR HTLA-ER NB cells, an SI value >> 10 was also observed, thus establishing the fact that the BPPB developed here can be considered safe for RBCs and promising for future developments for clinical use.

## 4. Conclusions

In this study, a bola-amphiphile molecule (BPPB) synthesized by a simple, low-cost, and one-step reaction has been assayed for the first time as an antitumor agent. Since bola-amphiphiles have never been investigated as possible anticancer agents, we thought carrying out in vitro experiments before using animal models would be rational. Structurally speaking, BPPB bears two identical triphenyl phosphonium (TPP) groups linked by a C12 alkyl chain conferring to BPPB nonpareil colloidal properties, allowing the self-assembly of nanovesicles of 49 nm. In particular, BPPB demonstrated very potent antiproliferative effects against neuroblastoma (NB) cells, including both sensitive HTLA-230 NB cells (IC_50_ = 0.2 µM) and multidrug-resistant HTLA-ER NB cells (IC_50_ = 1.1 µM), which have developed tolerance to ETO (IC_50_ = 592 µM), DOX, carboplatin, cisplatin, vincristine, and other clinically used drugs. In this regard, upon further necessary investigations with favorable outcomes on proper animal models, BPPB could be useful to treat HR-NB cells on which available therapies no longer function. Although its mechanism of action needs to be validated with further studies, which will be the topic of our next work, according to what is reported in the literature and new finding by us on other cancer cells, we can speculate that the TPP groups of BPPB could act first as cytoplasmic membrane disruptors, thus priming cell death. Then, TTP may redirect BPPB inside the mitochondria, where, upon accumulation by 100–1000 times, it could trigger various mechanisms of programmed death and apoptosis, impairing mitochondrial functions and causing oxidative stress (OS) by inducing ROS hyperproduction, mitochondrial toxicity, and inhibiting cellular respiration. Following these events, cell health would be irreversibly compromised, resulting in extensive death, by an extra-genomic mechanism, which can hardly determine the emergence of resistance. The anticancer activities of BPPB nanovesicles have been compared with those, which are already very good, of other previously reported nanosized compounds, establishing a striking superiority of BPPB in all cases. Even more importantly, the value of BPPB IC_50_ on HTLA-ER (IC_50_ = 1.1 µM) HR-NB cells was 538-fold lower than that reported for ETO (IC_50_ = 592 µM), a clinically approved drug to treat HR-NB cells, which unfortunately develops resistance after only six months of treatment. In addition, with the aim of developing a new anticancer agent potentially suitable for future clinical application, its pharmacological effects on non-tumoral cell lines, including cells of organs responsible for the detoxification of drugs such as liver and kidney, as well as red blood cells (RBCs), have been evaluated. Notably, the cytotoxicity of BPPB was investigated for the first time in vitro on human lung fibroblasts (MRC-5) by the Presto Blue ^TM^ viability test and on red blood cells (RBCs) by carrying out a reported protocol to evaluate the hemolytic ratio percentage. Results from these new investigations and those obtained on immortalized liver cells HepG2 and monkey kidney cells Cos-7 by carrying out MTT and LHD assays established that BPPB could display, at least in vitro, safe behavior on all cells considered here and especially toward RBCs. Specifically, after 24 h treatment, both NB cell populations were fully exterminated (14–18% of viable cells) at concentrations that left more than 98% of red blood cells alive. Although apparently not exceptional, the SI of BPPB for MDR HTLA-ER NB cells in relation to its cytotoxicity determined on the most susceptible non-tumoral human cells tested here (MRC-5) was 7.8-fold higher than that of ETO (0.32), as calculated using the reported average IC_50_ value obtained by Iijima et al., from evaluations on three human normal oral cell lines and the ETO IC_50_ value determined by us on the same NB cells after 24 h of exposure. Additionally, considering specifically the results obtained on HepG2 cells, which are a better in vitro model for the human organ responsible for drug detoxification, very high selectivity indices (SIs) were obtained against both HTLA-230 (SI = 40.6) and HTLA-ER (SI = 8.7). Also, this study confirmed that hindered bola-amphiphile TPP-based nanovesicles have high and selective biological effects, as often observed for several cationic nanomaterials. Collectively, at least in vitro, BPPB could therefore represent a novel potent template molecule to develop new clinically applicable anticancer agents for counteracting high-risk neuroblastoma. Future improvements in the anticancer effects of BPPB, while further reducing its cytotoxic profiles, could be derived by varying the length of the alkyl chain that acts as a linker between the two cationic heads and therefore by the liposomal formulation of the most promising compounds using themselves as cationic lipids.

## Data Availability

The original contributions presented in the study are included in the article/Appendix A, further inquiries can be directed to the corresponding authors.

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
