# Peer review of "The Remarkable and Selective In Vitro Cytotoxicity of Synthesized Bola-Amphiphilic Nanovesicles on Etoposide-Sensitive and -Resistant Neuroblastoma Cells"

_nanomaterials, 2024, doi:10.3390/nano14181505_

Round 1

Reviewer 1 Report

Comments and Suggestions for Authors

Authors have done comprehensive works on synthesis and toxicity test of 

Abstracton TPP- based bola amphiphiles NPs

There is no synthesis, characteristic methods and results in abstract.

Please include keywords of methods and some important data from your results in the abstract.

Section 3.1.1 

There is a huge table covering two pages. 

Please consider to select them and put a few figures.

Discussion

Lease compare result of characteristics 3.1.1 to literature.

Please include more information from introduction. There is cap between introduction and discussion.

Author Response

Authors have done comprehensive works on synthesis and toxicity test of

Abstracton TPP- based bola amphiphiles NPs

We apologize to the Reviewer for our ignorance, but what does he/she means by "Abstracton". Sorry, but not understanding and not finding the meaning of this word, we are unable to fully understand his/her comment.

There is no synthesis, characteristic methods and results in abstract.

Please include keywords of methods and some important data from your results in the abstract.

            If the Reviewer refers also to the synthesis and characterization of the compound BPPB, as already reported in the original main text (lines 242-245 and 360-363), they were described in our previous work, where its potent antibacterial effect has been testified. Such work was already cited in the not revised manuscript as preprint (Ref 35. Silvana Alfei; Guendalina Zuccari; Francesca Bacchetti; Carola Torazza; Marco Milanese; Carlo Siciliano; Constantinos M. Athanassopoulos; Gabriella Piatti; Anna Maria Schito Synthesized Bis-Triphenyl Phosphonium-Based Nano Vesicles Have Potent and Selective Antibacterial Effects on Several Clinically Relevant Superbugs. Preprint 2024, 2024070891, https://doi.org/10.20944/preprints202407.0891.v1), since still under review. Now, the work has been published and reference 35 has been updated, in the references list. So, to report in the abstract, as well as in the keywords, methods and results about the synthesis of BPPB appeared redundant to us. However, to satisfy the Reviewer, further specifications on this question have been included in the abstract. Please, check the modified abstract, expecially line 31. Concerning other important achievement belonging to this present work, as required by the Reviewer, they have been included in the abstract. Please, consider lines 36-48.

Section 3.1.1

There is a huge table covering two pages.

Please consider to select them and put a few figures.

On suggestion of the Reviewer, so as not to make the work more burdensome, “the huge Table covering two pages” in Section 3.1.1. (Table 1) has been moved in the Supplementary Materials file, where now it appears as Figure S1. Anyway, Table 1 (now Table S1, in the Supplementary Materials) has been only slightly modified, because it reports essential data for readers to have clear the physicochemical characteristics of BPPB and its biological effects previously reported. The content of the Supplementary Materials file at the end of the manuscript has been updated (lines 914-915) as the Tables numbering.

Discussion

Lease compare result of characteristics 3.1.1 to literature.

The requested comparison has been already reported in our previous work (here cited as Ref. 35), in which the synthesis and complete characterization of BPPB has been described and our results were compared with those of Ceccacci et al. (also cited in the present work as Ref. 29). To repeat such comparison appeared us redundant and out of the scope of the present work.

Please include more information from introduction. There is cap between introduction and discussion.

 We apologize to the Reviewer for our ignorance, but what does he/she means by "Cap"? Sorry, but not understanding the meaning of this word, we are unable to fully understand his/her comment. Anyway, if he/she intended “gap”, we kindly make note that “Introduction” and “Discussion” should not be interconnected but should be two independent Sections. The first should provide an as complete as possible background to locate the work in a specific sector followed by an explanation of the scopes of the study (and our Introduction possesses such requirements), while the second (discussion) should provide a discussion/comments/analyses of the results and findings achieved. The gap is even necessary. Please, consider the Instructions of Nanomaterials concerning Discussion Section:

  • Discussion: Authors should discuss the results and how they can be interpreted in perspective of previous studies and of the working hypotheses. The findings and their implications should be discussed in the broadest context possible, and limitations of the work highlighted. Future research directions may also be mentioned. This section may be combined with Results.

Reviewer 2 Report

Comments and Suggestions for Authors

In this manuscript, authors synthesizes a molecule BPPB that demonstrated very potent antiproliferative effects against neuroblastoma (NB) cells, including both sensitive HTLA-230 NB cells (IC50 = 0.2 µM) and multidrug resistant HTLA-ER NB cells (IC50 = 1.1 µM). Compared with COS-7 and HepG2, BPPB does selectively kill NB cells after 24 hours treatments, however, it has no obvious selectivity for MRC-5 cells or for longer times of treatment, so concentration-dependent cytotoxicity of BPPB is worrying. In addition, this study only carried out in vitro cell experiments, without animal experiments, so the current research results can not be published. In addition, there are some questions that need author to pay attention to.

1, The abstract is a bit miscellaneous, the focus is not prominent enough, it is suggested to add some quantitative description, extract the main points to reduce the content of the abstract.

2, Too many keywords can be appropriately deleted, retain the main keywords, suggest five.

3, Although some possible mechanisms, such as oxidative stress and mitochondrial targeting, are mentioned, the specific roles and effects of these mechanisms are not explained and discussed in detail.

4, For the cytotoxicity of the drug, it is recommended to increase animal experiments to determine the cytotoxicity.

5, The Table 1 on page 10 and 11 is not clear and looks confusing. It is recommended to separate pictures and data, and to enlarge the font and make it black.

Comments on the Quality of English Language

  The quality of English language is good, and the writing of the article needs improvement.

Author Response

In this manuscript, authors synthesizes a molecule BPPB that demonstrated very potent antiproliferative effects against neuroblastoma (NB) cells, including both sensitive HTLA-230 NB cells (IC50 = 0.2 µM) and multidrug resistant HTLA-ER NB cells (IC50 = 1.1 µM). Compared with COS-7 and HepG2, BPPB does selectively kill NB cells after 24 hours treatments, however, it has no obvious selectivity for MRC-5 cells or for longer times of treatment, so concentration-dependent cytotoxicity of BPPB is worrying.

We thank the Reviewer for his/her comment that offered us the possibility to better clarify him/her the acceptable to high safety (at least in vitro) of BPPB, when administered to inhibit NB cells. As reported in the main text and observable in the original Table 3 (Table 2 in the revised manuscript), the IC50 values on both NB cell populations after the first 24 hours of treatment, as well as cell viability profiles, did not differ significantly from those measured for longer exposure timing (48 and 72 hours). So, we decided to consider 24 hours as the maximum time of a possible treatment using BPPB, since, after such time, NB cells are already fully exterminated. With such treatment, as reported in Table 6 (now Table 5), also considering MRC-5 cells, the SI values were significantly higher than 1 (please, consider the reference 68 by Krzywik et al, cited in the text), thus establishing a good selectivity for both sensitive (SI = 11.7) and MDR (SI = 2.5) NB cells. Also, the Reviewer should consider that lung fibroblast used in this study, in addition to not being tumoral cells, are also not immortalized cells, and therefore more sensitive to the effects of drugs.

In this regard, a discussion on the differences in the susceptibility of the different types of not tumoral cells chosen by us, had been already included in the original main text.

Even more important, the relevance of our results (at least in vitro) derives by the very potent cytotoxicity of BPPB to NB cells, also if compared to that (in vitro) of etoposide (ETO), which is one of the most clinically used drug to treat HR-NB. As reported, in vitro results demonstrated that upon administration of 100 µM etoposide for 24 hours, i.e. a dose 91 and 500-times higher than that needed of BPPB to kill the 50% of NB cells, cell viability of HTLA-ER and HTLA-230 remained over 80% and close to 70% respectively (https://doi.org/10.18632/oncotarget.12209). Also, in a more recent work, it was reported that the extrapolated IC50 of ETO on HTLA-ER was 592 µM (https://doi.org/10.3390/ijms241915027), i.e. 538-fold higher than that of BPPB (1.1 µM). Moreover, the apparently not exceptional SI of BPPB for MDR HTLA-ER NB cells, calculated considering its cytotoxicity vs. MRC-5 cells after 24 hour treatment, was remarkably higher than that of ETO, calculated using the reported average IC50 value obtained by Iijima et al, from evaluations on three human normal oral cell lines [Ref. 69] and the ETO IC50 value determined by us on the same NB cells, after 24 hours exposure [Ref. 58]. Specifically, the SI of BPPB for MDR HTLA-ER NB cells (2.49) was 7.8-fold higher than that of ETO (0.32). On these discoveries, between those of BPPB and those of ETO (clinically approved), which concentrations/dependent antitumor effects of the active principles could be less worrying and safer on MRC-5 cells, for the Reviewer?

These comments and data not reported previously, and other specifications have been now included in the main text, at lines 36-38, 523-530, 808-815, 865, 880-883 and 895-900.

In addition, this study only carried out in vitro cell experiments, without animal experiments, so the current research results can not be published.

We apologise in advance with the Reviewer, but his/her assertion has not robust foundation. The literature is full of studies that report only in vitro biological experiments and not investigations on animals, that are required in more advanced stages of research. In fact, from a quick survey on PubChem database using “Studies in vitro against neuroblastoma” and “Studies in vivo against neuroblastoma” the results in the years 1980-2024 were 1,046 and only 540, respectively. So, why our current in vitro research results cannot be published? Experimentations on animals were not within the scope of the work nor make they part of the research typology of our group. Considering that bola-amphiphilic molecules were never investigated as anticancer agents so far, we thought that an early extensive in vitro investigation was needed before considering scarifying animals. We kindly ask the Reviewer to contemplate also, the extensive in vitro study that we have performed even if in vitro: we have tested our compound on two tumoral cell populations, one of which is MDR and on 4 non-tumoral cell lines to investigate the possible and future clinical applicability of BPPB and if it could worthy of further studies including those on animals. We therefore friendly request the Reviewer to be satisfied with the in vitro results reported here that are truly exceptional and robust. However, to avoid misunderstandings and disappointments in readers like those of the Reviewer, the fact that the results reported concern only in vitro experiments has been pointed out from the title. In addition to those already present in the original manuscript several further “in vitro” have been included in the revised manuscript. Finally, supplementary specification on the question and on the scopes of the present study, have been included in lines 46-48, 226-228, 233-234, 857-859 and 866-867.

In addition, there are some questions that need author to pay attention to.

1, The abstract is a bit miscellaneous, the focus is not prominent enough, it is suggested to add some quantitative description, extract the main points to reduce the content of the abstract.

We thank the Reviewer for his/her suggestion which enabled us to improve the Abstract content. The Abstract has been modified. Please, check it.

2, Too many keywords can be appropriately deleted, retain the main keywords, suggest five.

Although Nanomaterials allowed up to ten keywords, we have reduced them to six as asked by the Reviewer.

3, Although some possible mechanisms, such as oxidative stress and mitochondrial targeting, are mentioned, the specific roles and effects of these mechanisms are not explained and discussed in detail.

We thank the Reviewer for his/her comments. Additional details on the assumed mechanism of action of BPPB and new findings due to an ongoing study on BPPB have been included in the text which could confirm the reported hypothesis based on literature reports. Please, see lines 530-546.

4, For the cytotoxicity of the drug, it is recommended to increase animal experiments to determine the cytotoxicity.

Our answer to this point has been already included in our replay to a previous point. Please, reconsider the part evidenced in yellow.

5, The Table 1 on page 10 and 11 is not clear and looks confusing. It is recommended to separate pictures and data, and to enlarge the font and make it black.

We thank the Reviewer for his/her suggestion which enabled us to improve the quality of Table 1, which has been modified following some of the Reviewer indications. Also, to satisfy another Reviewer, Table 1 has been moved in the Supplementary Materials file, where now it appears as Table S1. The content of the Supplementary Materials file at the end of the manuscript has been updated (lines 914-915) as the Tables numbering.

Comments on the Quality of English Language

The quality of English language is good, and the writing of the article needs improvement.

We thank the Reviewer for his/her positive comment on the English language. Anyway, with the help of our colleague prof Deirdre Kantz, teacher mother tongue of English at University of Genoa and Pavia, the work has been revised to further reduce typos and grammatical errors.

Reviewer 3 Report

Comments and Suggestions for Authors

The paper entitiled “Remarkable and Selective Cytotoxicity of Synthesized Bola- 2 Amphiphilic Nano Vesicles on Etoposide-Sensitive and Resistant Neuroblastoma Cells” deals with biological activity of bis-cationic bilo-amphiphilic BPPB (formula at Fig 1). It was tested on several cell lines in vitro, with special attention to high-risk neuroblastoma model lines: HTLA-ER and HTLA-230.

In the excellent Introduction Authors explain why resistant forms of NB were chosen, together with human lung fibroblast MRC-5, immortalized human liver cells HepG2, and monkey kidney fibroblast Cos-7. All cell lines were subjected to time- and dose-dependent exposure to BPPB.

The BPPB was synthesized previously and brief characterization of this compound is presented in the main text, as well as in Supplementary Materials. The compound is water soluble, although it associates to give ca 50 nm diameter nanoparticles, with positive zeta potential, ca 18 mV. Accordingly BPPB penetrates negatively charged cell and mitochondrial barriers, which  is promising for hypothetic anticancer activity.

The results on IC-50 against both NB lines are low (much below 1 microM concentration) in comparison with IC-50 against HepG2 and Cos-7, which were at least one order of magnitude resistant to BPPB. MRC-5 lines were more susceptible to BPPB within 0.5-5 microM concentration and showed IC-50 ca 1 microM in 48 and/or 72 hr test.

Additional hemolytic toxicity was also tested on RB cells within 0.1-50 microM concentration of BPPB. The results showed that BPPB s innocent up to 5 microM.

High selectivity indices (SI) calculated for BPPB towards NB lines vs reference lines MRC-5, HepG2 and Cos-7 in case of HTLS-230 line are striking, while SI vs RB were above 10, although again in case of HTLS-230 line SI was 75.

All the results allow to conclude that BPPB could be useful to treat HR-NB cells on which available therapies are no longer functioning.

It is very well written paper, so nicely presented although very long. Supplementary materials are very useful.

What I found, which can be improved is:

1.Text corrections remained visible: lines 132, 144, 258, 665; please remove unnecessary words, which are already crossed out.

2.In Supplementary Materials the character “micro” in units on horizontal axes needs improvement: Fig.S1-S7, S12-17, and vertical S18.

The paper is sounding ! I have learned a lot reading it, thank you

Comments on the Quality of English Language

Very well written

Author Response

The paper entitiled “Remarkable and Selective Cytotoxicity of Synthesized Bola- 2 Amphiphilic Nano Vesicles on Etoposide-Sensitive and Resistant Neuroblastoma Cells” deals with biological activity of bis-cationic bilo-amphiphilic BPPB (formula at Fig 1). It was tested on several cell lines in vitro, with special attention to high-risk neuroblastoma model lines: HTLA-ER and HTLA-230.

In the excellent Introduction Authors explain why resistant forms of NB were chosen, together with human lung fibroblast MRC-5, immortalized human liver cells HepG2, and monkey kidney fibroblast Cos-7. All cell lines were subjected to time- and dose-dependent exposure to BPPB.

We thank a lot the Reviewer for his/her positive comments and appreciations.

The BPPB was synthesized previously and brief characterization of this compound is presented in the main text, as well as in Supplementary Materials. The compound is water soluble, although it associates to give ca 50 nm diameter nanoparticles, with positive zeta potential, ca 18 mV. Accordingly BPPB penetrates negatively charged cell and mitochondrial barriers, which is promising for hypothetic anticancer activity.

The results on IC-50 against both NB lines are low (much below 1 microM concentration) in comparison with IC-50 against HepG2 and Cos-7, which were at least one order of magnitude resistant to BPPB. MRC-5 lines were more susceptible to BPPB within 0.5-5 microM concentration and showed IC-50 ca 1 microM in 48 and/or 72 hr test.

Additional hemolytic toxicity was also tested on RB cells within 0.1-50 microM concentration of BPPB. The results showed that BPPB s innocent up to 5 microM.

High selectivity indices (SI) calculated for BPPB towards NB lines vs reference lines MRC-5, HepG2 and Cos-7 in case of HTLS-230 line are striking, while SI vs RB were above 10, although again in case of HTLS-230 line SI was 75.

All the results allow to conclude that BPPB could be useful to treat HR-NB cells on which available therapies are no longer functioning.

We thank a lot the Reviewer for his/her interest in our study and findings.

It is very well written paper, so nicely presented although very long. Supplementary materials are very useful.

We thank a lot the Reviewer for his/her positive comments and appreciations.

What I found, which can be improved is:

1.Text corrections remained visible: lines 132, 144, 258, 665; please remove unnecessary words, which are already crossed out.

We thank a lot the Reviewer for his/her indications. Sorry, the issue has been addressed.

2.In Supplementary Materials the character “micro” in units on horizontal axes needs improvement: Fig.S1-S7, S12-17, and vertical S18.

We apologize in advance to the Reviewer, but the character is that provided by the software and cannot be modified. We ask kindly the Reviewer to accept such character although not very nice.

The paper is sounding! I have learned a lot reading it, thank you

We thank a lot the Reviewer for his/her positive comments and appreciations.

Comments on the Quality of English Language

Very well written

We thank a lot the Reviewer for his/her positive comments and appreciations for our English language.

Round 2

Reviewer 1 Report

Comments and Suggestions for Authors

Authors have improved manuscript.

Methodology

The Graphpad software has been included. Please include which type of statistical analysis.

Result

Figure 7,8,9,10

Please write µM at X axle in the graph.

Please include scale bar in panels of figure S9 and S10

Author Response

Authors have improved manuscript.

Methodology

The Graphpad software has been included. Please include which type of statistical analysis.

We apologize to the Reviewer in advance, but we make kindly note that the information he/she requires i.e.  “which type of statistical analysis” was already included in the first version of unmodified manuscript in Section 2.2.4., and in the captions o Figure 2, 3, 4, 5, 7, 8, 9 and 10.

Result

Figure 7,8,9,10

Please write µM at X axle in the graph.

We apologize to the Reviewer in advance, but we make kindly note that the information he/she requires was already included in the first version of unmodified manuscript. Specifically, in Figure 7 the measure unit is inserted in the x axis title, in Figure 8 it is inserted close to all numbers on x axis, in Figure 9 it is included both in the x axis title, and close to all numbers on x axis, and in Figure 10 it is inserted in the x axis title as in Figure 7.

Please include scale bar in panels of figure S9 and S10

We apologize to the Reviewer in advance, but we make kindly note that specifications about the information he/she requires were already included in the first version of unmodified manuscript, in the caption related to each Figure. Specifically, in Figure S8 caption, there is a complete description of panels content, including the scale bars. Consequently, since no change occurred in Figure S9 and S10, all this information has not been repeated and the sentence:” Iconography as for Figure S8”, was inserted. Is it still not clear?